# LEARNING A UNIFIED LABEL SPACE

## ABSTRACT

How do we build a general and broad object detection system? We use all labels of all concepts ever annotated. These labels span many diverse datasets with potentially inconsistent semantic labels. In this paper, we show how to integrate these datasets and their semantic taxonomies in a completely automated fashion. Once integrated, we train an off-the-shelf object detector on the union of the datasets. This unified recognition system performs as well as dataset-specific models on each training domain, but generalizes much better to new unseen domains. Entries based on the presented methodology ranked first in the object detection and instance segmentation tracks of the ECCV 2020 Robust Vision Challenge.

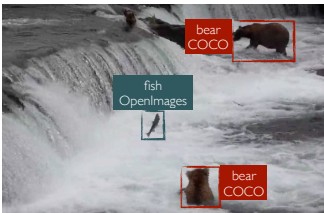 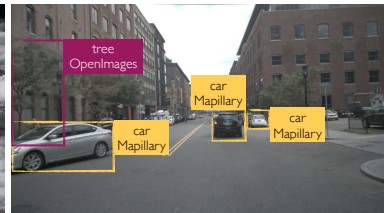 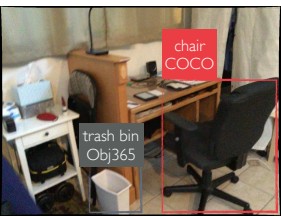

Figure 1: Different datasets span diverse semantic and visual domains. We learn to unify the label spaces of multiple datasets and train a single object detector that generalizes across datasets.

## 1 INTRODUCTION

Computer vision aims to produce broad, general-purpose perception systems that work in the wild. Yet object detection is fragmented into datasets (Lin et al., 2014; Neuhold et al., 2017; Shao et al., 2019; Kuznetsova et al., 2020) and our models are locked into specific domains. This fragmentation brought rapid progress in object detection (Ren et al., 2015) and instance segmentation (He et al., 2017), but comes with a drawback. Single datasets are limited and do not yield general-purpose recognition systems. Can we alleviate these limitations by unifying diverse detection datasets?

In this paper, we make training an object detector on the union of disparate datasets as straightforward as training on a single one. The core challenge lies in integrating different datasets into a common taxonomy and label space. A traditional approach is to create this taxonomy by hand (Lambert et al., 2020; Zhao et al., 2020), which is both time-consuming and error-prone. We present a fully automatic way to unify the output space of a multi-dataset detection system using visual data only. We use the fact that object detectors for similar concepts from different datasets fire on similar novel objects. This allows us to define the cost of merging concepts across datasets, and optimize for a common taxonomy fully automatically. Our optimization jointly finds a unified taxonomy, a mapping from this taxonomy to each dataset, and a detector over the unified taxonomy using a novel 0-1 integer programming formulation. An object detector trained on this unified taxonomy has a large, automatically constructed vocabulary of concepts from all training datasets.

We evaluate our unified object detector at an unprecedented scale. We train a unified detector on 4 large and diverse datasets: COCO (Lin et al., 2014), Objects365 (Shao et al., 2019), OpenImages (Kuznetsova et al., 2020), and Mapillary (Neuhold et al., 2017). Experiments show that our learned taxonomy outperforms the best expert-annotated label spaces, as well as language-based alternatives. For the first time, we show that a single detector performs as well as dataset-specific models on each individual dataset. Crucially, we show that models trained on the diverse training sets generalize zero-shot to new domains, and outperform single-dataset models. Our models ranked first in the object detection and instance segmentation tracks of the ECCV 2020 Robust Vision Challenge across all evaluation datasets. Code and models will be released upon acceptance.

## 2 Related Work

**Training on multiple datasets.** In recent years, training on multiple diverse datasets has emerged as an effective tool to improve model robustness for depth estimation (Ranftl et al., 2020) and stereo matching (Yang et al., 2019). In these domains unifying the output space involves modeling different camera models and depth ambiguities. In contrast, for recognition, the unification involves merging different semantic concepts. MSeg (Lambert et al., 2020) manually created a unified label taxonomy of 7 semantic segmentation datasets and used Amazon Mechanical Turk to resolve the inconsistent annotations between datasets. Different from MSeg, our solution does not require any manual effort and unifies the label space directly from visual data in a fully automatic way.

Wang et al. (2019) train a universal object detector on multiple datasets, and gain robustness by joining diverse sources of supervision. However, they produce a dataset-specific prediction for each input image. When evaluated in-domain, they require knowledge of the test domain. When evaluated out-of-domain, they produce multiple outputs for a single concept. This limits the generalization ability of detection, as we show in experiments (Section 5.2). Our approach, on the other hand, merges visual concepts at training time and yields a single consistent model that does not require knowledge of the test domain and can be deployed cleanly in new domains. Both Wang et al. (2019) and MSeg (Lambert et al., 2020) observe a performance drop in a single unified model. With our unified label space and a dedicated training framework, this is not the case: the unified model performs as well as single-dataset models on the training datasets.

Zhao et al. (2020) trains a universal detector on multiple datasets: COCO (Lin et al., 2014), Pascal VOC (Everingham et al., 2010), and SUN-RGBD (Song et al., 2015), with under 100 classes in total. They manually merge the taxonomies and then train with cross-dataset pseudo-labels generated by dataset-specific models. The pseudo-label idea is complementary to our work. Our unified label space learning removes the manual labor, and works on a much larger scale: we unify COCO, Objects365, and OpenImages, with more complex label spaces and $900+$ classes.

YOLO9000 (Redmon & Farhadi, 2017) combines detection and classification datasets to expand the detection vocabulary. LVIS (Gupta et al., 2019) extents COCO annotations to $> 1000$ classes in a federated way. Our approach of fusing multiple readily annotated datasets is complementary and can be operationalized with no manual effort to unify disparate object detection datasets.

**Zero-shot classification and detection** reason about novel object categories outside the training set (Fu et al., 2018; Bansal et al., 2018). This is often realized by representing a novel class by a semantic embedding (Norouzi et al., 2014) or auxiliary attribute annotations (Farhadi et al., 2009). In zero-shot detection, Bansal et al. (2018) proposed a statically assigned background model to avoid novel classes being detected as background. Rahman et al. (2019) included the novel class word embedding in test-time training to progressively generate novel class labels. Li et al. (2019) leveraged external text descriptions for novel objects. Our program is complementary: we aim to build a sufficiently large label space by merging diverse detection datasets during training, such that the trained detector transfers well across domains even without machinery such as word embeddings or attributes. Such machinery can be added, if desired, to further expand the model's vocabulary.

## 3 Preliminaries

An object detector jointly predicts the locations $b_k \in \mathbb{R}^4$ and classwise detection scores $d_k \in \mathbb{R}^{|L|}$ of all objects in a scene. The detection score describes the confidence that a bounding box belongs to an object with label $l \in L$, where $L$ is the set of all classes. Figure 2a provides an overview. On a single dataset, the detector is trained to produce high scores only for the ground-truth class.

Consider multiple datasets, each with its own label space $\hat{L}^1, \hat{L}^2, \ldots$. A detector now needs to learn a common label space $L$ for all datasets, and define a mapping between this common label space and dataset-specific labels $L \rightarrow \hat{L}^i$. In this work, we only consider direct mappings. Each common label maps to at most one dataset-specific label per dataset, and each dataset-specific label maps to exactly one common label. In particular, we do not hierarchically relate concepts across datasets. When there are different label granularities between datasets, we keep them all in our label space, and expect to predict all of them. Mathematically, the mapping from the joint output space to a dataset-specific one is a Boolean linear transformation of the output of the recognition system

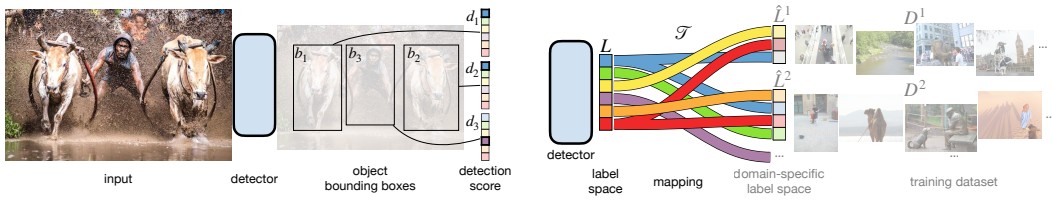

(a) Object detection pipeline  (b) Our training framework

Figure 2: A standard object detection pipeline (a) fits bounding boxes to objects and predicts detection scores over a fixed set of output classes. In single-dataset training, class scores are supervised directly from annotations in the dataset. In multi-dataset training (b), a detector learns its own output space and links it to the label space of each dataset.

$\hat{d}_k^i = \mathcal{T}^i d_k$, with $\hat{d}_k^i \in \mathbb{R}^{|\hat{L}_i|}$, $\mathcal{T}^i \in \{0,1\}^{|\hat{L}^i| \times |L|}$, and constraints $\mathcal{T}^i \mathbf{1} = \mathbf{1}$, $\mathcal{T}^{i\top} \mathbf{1} \le \mathbf{1}$. The two constraints ensure that only direct mappings are learned. For simplicity, let $\mathcal{T}^\top = \left[\mathcal{T}^{1\top} \dots, \mathcal{T}^{N\top}\right]$ be the mapping to all dataset-specific output spaces. Figure 2b provides an overview.

Prior work defined $L$ and $\mathcal{T}$ by hand (Lambert et al., 2020; Zhao et al., 2020) or used a trivial mapping $\mathcal{T} = I$ with completely disjoint outputs (Wang et al., 2019). They then trained a detector given the fixed label space and mapping. In the next section, we show how to jointly learn the label space, the mapping to the individual datasets, and the detection scores in a globally optimal manner.

## 4 METHOD

We start with training a detector on the trivial disjoint label space $\bigcup_k \hat{L}^k$. In this section, we show how to automatically learn a unified label space by converting the disjoint label space into a unified label space. Once the unified label space is learned, we retrain the detector end-to-end with the unified label space. An overview of our workflow can be found in Appendix G

### 4.1 LEARNING A UNIFIED LABEL SPACE

We first consider only fine-tuning the last linear layer of the disjoint-label space detectpr. Specifically, let $f_1, f_2, \dots$ be the $D$-dimensional features $f_i \in \mathbb{R}^D$ of the penultimate layer of the pretrained model for object locations $b_1, b_2, \dots$ for all objects in a dataset. Our goal is to learn a new detection score $d_k = W^\top f_k$ with parameters $W = \left[w_1, w_2, \dots, w_{|L|}\right]$ and $w_l \in \mathbb{R}^D$, a label space $L$, and dataset-specific transformations $\mathcal{T}$. The pretrained detector allows us to formulate this objective over a fixed set of precomputed detections and their features $F = [f_1, f_2, \dots]$:

$$\text{minimize}_{L, \mathcal{T}, W} \qquad \sum_{l \in L} \ell_l(\mathcal{T}_l w_l^\top F) + \lambda |L| \tag{1}$$

$$\text{subject to} \qquad \mathcal{T}^i \mathbf{1} = \mathbf{1} \quad \text{and} \quad \mathcal{T}^{i\top} \mathbf{1} \le \mathbf{1} \qquad \forall_{i \in \{1 \dots N\}}.$$

Here $\ell_l$ is a general loss function that factorizes over the labels $l \in L$, and $N$ is the number of datasets. The weight $w_l$ controls the output of the detector for the joint label $l$, and $\mathcal{T}_l$ is a column of the dataset-specific transformation that maps each joint label $l$ to all training datasets. The cardinality penalty $\lambda |L|$ encourages a small and compact label set. A factorization of the loss $\ell_l$ over the output space $l \in L$ may seem restrictive. However, it does include the most common loss functions in detection: sigmoid cross-entropy and mean average precision. Section 4.2 discusses the exact loss functions used in our optimization.

For a fixed label set $L$ and mapping $\mathcal{T}$, objective 1 reduces to a standard training objective of a detector. However, the joint optimization of $L$ and $\mathcal{T}$ significantly complicates the optimization. It mixes combinatorial optimization over $L$ with continuous optimization of $W$, and a 0-1 integer program over $\mathcal{T}$. However, there is a simple reparametrization that lends itself to efficient optimization.

First, observe that the label set $L$ simply corresponds to the number of columns in $\mathcal{T}$. Furthermore, we merge at most one label per dataset $\mathcal{T}^{i\top} \mathbf{1} \le \mathbf{1}$. Hence, for each dataset $i$ a column $\mathcal{T}_l^i \in \mathbb{T}^i$ takes one of $|\hat{L}_i| + 1$ values: $\mathbb{T}^i = \{\mathbf{0}, 1_{\hat{l}_1}, 1_{\hat{l}_2}, \dots\}$, where $1_{\hat{l}} \in \{0, 1\}^{|\hat{L}_i|}$ is an indicator vector. Each column $\mathcal{T}_l \in \mathbb{T}$ then only chooses from a small set of potential values $\mathbb{T} = \mathbb{T}^1 \times \mathbb{T}^2 \times \dots$, where $\times$ represents the Cartesian product. Instead of optimizing over the label set $L$ and transformation $\mathcal{T}$

directly, we instead use combinatorial optimization over the potential column values of $t \in \mathbb{T}$. Let $x_t \in \{0, 1\}$ be the indicator of combination $t \in \mathbb{T}$. In this combinatorial formulation, the constraint $\mathcal{T}^i \mathbf{1} = \mathbf{1}$ translates to $\sum_{t \in \mathbb{T}|t_{\hat{l}}=1} x_t = 1$ for all dataset-specific labels $\hat{l}$. Furthermore, the objective of the optimization simplifies to

$$\sum_{l \in L} \ell_l(\mathcal{T}_l w_l^\top F) + \lambda|L| = \sum_{t \in \mathbb{T}} x_t \ell_l(t w_t^\top F) + \lambda \sum_{t \in \mathbb{T}} x_t. \tag{2}$$

Crucially, the weights $w_t$ of the detection score are now independent of the combinatorial optimization, and can be precomputed for each column value $t \in \mathbb{T}$ in a merge cost:

$$c_t = \min_{w_t} \ell_t(t w_t^\top F). \tag{3}$$

This leads to a compact integer linear programming formulation of objective 1:

$$\text{minimize}_x \qquad \sum_{t \in \mathbb{T}} x_t \left(c_t + \lambda\right)$$
$$\text{subject to} \qquad \sum_{t \in \mathbb{T}|t_{\hat{l}}=1} x_t = 1 \qquad \forall_{\hat{l}} \tag{4}$$

For two datasets, the above objective is equivalent to a weighted bipartite matching. For a higher number of datasets, it reduces to weighted graph matching and is NP-hard, but is practically solvable with integer linear programming (Linderoth & Ralphs, 2005).

One of the most appealing properties of the above formulation is that it separates the integer optimization over $x_t$ from the continuous optimization of the last linear layer $w_t$. Section 4.2 explores this separation and shows how to precompute the merge cost $c_t$ for various loss functions.

One major drawback of the combinatorial reformulation is that the set of potential combinations $\mathbb{T}$ grows exponentially in the number of datasets used: $|\mathbb{T}| = O(|\hat{L}_1||\hat{L}_2||\hat{L}_3|\ldots)$. However, most merges $t \in \mathbb{T}$ are arbitrary combinations of labels and incur a large merge cost $c_t$. Section 4.3 presents a linear-time greedy enumeration algorithm for low-cost merges.

Considering only low-cost matches, standard integer linear programming solvers find an optimal solution within a second for all label spaces we tried, even for $|L| > 600$ and up to 6 datasets.

## 4.2 Loss functions

The loss function in our constrained objective 1 is quite general and captures a wide range of commonly used losses. We highlight two: an unsupervised objective based on the distortion of the output compared to the pretrained model, and mean Average Precision (mAP).

**Distortion.** A natural objective is to learn a joint label space that stays close to the pretrained model. Let $\bar{D} = \bar{W}^\top F$ be the detection scores of the pretrained model with weights $\bar{W} = [\bar{w}_1, \bar{w}_2, \ldots]$ for each dataset-specific label. Distortion then measures the difference between the joint and pretrained models on all dataset-specific outputs:

$$\ell_t^{\text{dist}}(t w_t^\top F) = \frac{1}{2} \sum_{\hat{l}} t_{\hat{l}} \left(\bar{w}_{\hat{l}}^\top F - w_t^\top F\right)^2. \tag{5}$$

This distortion has a closed-form solution $w_t = \frac{\sum_{\hat{l}} t_{\hat{l}} \bar{w}_{\hat{l}}}{\sum_{\hat{l}} t_{\hat{l}}}$. The merge cost $c_t$ corresponds to the variance in detector outputs between dataset-specific labels and can be computed efficiently from pairwise differences. The main drawback of this distortion measure is that it does not take task performance into consideration when optimizing the joint label space. Next, we show how to learn a label space using annotations for dataset-specific predictions on the original datasets.

**Mean Average Precision.** Let $\text{mAP}_{\hat{l}}$ be the mean Average Precision for each dataset-specific class $\hat{l}$ on the corresponding dataset-specific validation set. Let $\text{mAP}_t$ be the mAP of a merged output over all dataset-specific outputs used in $t$. Our loss is then the difference in mAP:

$$\ell_t^{\text{mAP}}(t w_t^\top F) = \sum_{\hat{l}} (t_{\hat{l}} \text{mAP}_{\hat{l}} - \text{mAP}_t). \tag{6}$$

It is hard to optimize $w_t$ for mAP directly, since it operates on a sorted set of detection scores. We instead use the averaging solution of the distortion metric (5), and simply evaluate $c_t$. The mAP computation is computationally quite expensive, but we will provide an optimized joint evaluation with our code.

### 4.3 COMPUTATION AND PRUNING

The size of our optimization problem scales linearly in the number of potential merges $|\mathbb{T}|$, which can grow exponentially in the number of datasets. To counteract this exponential growth, we only consider sets of classes

$$\mathbb{T}' = \left\{ \boldsymbol{t} \in \mathbb{T} \,\middle|\, \frac{c_t}{|\boldsymbol{t}| - 1} \leq \tau \right\}.$$

For an aggressive enough threshold $\tau$, the number of potential merges $|\mathcal{T}'|$ remains manageable. We greedily grow $\mathcal{T}'$ by first enumerating all feasible two-class merges ($|\boldsymbol{t}| = 2$), then three-class merges, and so on. We use $\lambda = 0.1$ and $\tau = 0.2$ in our experiments. The runtime of this greedy algorithm is $O(|\mathcal{T}'| \max_i |\hat{L}^i|)$. In practice, the cost computation took a few seconds for the distortion loss function and about 10 minutes for the mAP loss (due to the need to repeatedly recompute mAP). The integer programming solver finds the optimal solution within one second in both cases.

## 5 EXPERIMENTS

Our goal aims to facilitate the training of models that perform well across datasets. Our main training datasets are adopted from the ECCV 2020 Robust Vision Challenge (RVC). These are four large object detection datasets: COCO (Lin et al., 2014), OpenImages (Kuznetsova et al., 2020), Objects365 (Shao et al., 2019), and Mapillary (Neuhold et al., 2017). To evaluate zero-shot cross-dataset generalization, we use the RVC instance segmentation datasets (all of which have bounding box annotations): VIPER (Richter et al., 2017), CityScapes (Cordts et al., 2016), ScanNet (Dai et al., 2017), WildDash (Zendel et al., 2018), and KITTI (Geiger et al., 2012). In addition, we test zero-shot on two object detection datasets: Pascal VOC (Everingham et al., 2010) and CrowdHuman (Shao et al., 2018). The datasets are described in more detail in Appendix A.

In our implementation, we first train a unified detector on the large and general datasets: COCO, Objects365, and OpenImages. As Mapillary is small and domain-specific, we add it in a subsequent fine-tuning stage. See Appendix B for details.

**Evaluation metric.** We evaluate a detector both on its training dataset(s) and new datasets that may contain objects out of the training label space. On the training datasets, we use the official metrics of each datasets: for COCO, Objects365, and Mapillary, we use the mAP at IoU thresholds 0.5 to 0.95. For OpenImages, we use the official modified mAP@0.5 that excludes unlabeled classes and enforces hierarchical labels (Kuznetsova et al., 2020).

For evaluating on new test datasets, standard mAP evaluation requires an expert-annotated test-to-train class correspondence. This is unavailable in our setting. We initially invited 5 volunteer annotators to link the classes between each test dataset and the unified label space. However, we observe considerable disagreement between annotators. For example, "rider" in CityScapes is linked to 'bicyclist' or 'motorcyclist' by different human annotators. For reproducible and scalable evaluation, we instead define a new metric, mean Expected AP (mEAP). We use a word embedding (Pennington et al., 2014) to find a set of correspondences between each test class and the unified label space. Each test class can have multiple correspondences. We calculate the expected AP (EAP) of a test class as the average AP of all corresponding unified classes. The summary metric per test dataset over all classes is mean EAP (mEAP). In most cases, a test class corresponds to only one joint label, and the EAP and AP are equivalent. Appendix C specifies the evaluation protocol in more detail.

### 5.1 IMPLEMENTATION

We use the CascadeRCNN detector (Cai & Vasconcelos, 2019). A single region proposal network (RPN) is shared by all datasets. Each proposed bounding box from RPN is classified by a cascaded classifier. For a disjoint label-space baseline, the last classification layer of each cascade stage is split between datasets. We only apply a training loss to the source classifier, following Wang et al. (2019). Our unified detector uses CascadeRCNN as is.

|  | VOC | VIPER | CityScapes | ScanNet | WildDash | CrowdH. | KITTI | *mean* |
|---|---|---|---|---|---|---|---|---|
| COCO | 80.0 | 13.9 | 39.6 | 17.4 | 25.9 | **73.9** | 30.5 | 40.2 |
| Objects365 | 71.9 | 20.7 | 43.4 | 24.9 | 27.6 | 71.8 | 32.2 | 41.8 |
| OpenImages | 64.4 | 10.4 | 29.8 | 24.2 | 20.3 | 66.7 | 21.8 | 33.9 |
| Mapillary | 11.4 | 15.2 | 44.7 | 0.0 | 23.4 | 49.3 | 37.8 | 26.0 |
| Disjoint label-space | 80.5 | 19.5 | 44.7 | **32.4** | 30.0 | 65.3 | 35.9 | 44.0 |
| Unified (ours) | **82.5** | **20.9** | **49.1** | 30.8 | **31.6** | 70.7 | **37.9** | **46.2** |
| Unified (expert human) | 82.5 | 20.9 | 50.3 | 31.4 | 31.9 | 71.3 | 37.1 | 46.5 |
| Dataset-specific | 80.3 | 31.8 | 54.6 | 44.7 | - | 80.0 | - | - |

Table 1: Zero-shot cross-dataset object detection performance on the validation sets of datasets that were not seen during training. The metric is mEAP@0.5 (see the text for a definition). We compare to models trained on each single training dataset (top 4 rows), a unified detector with dataset-specific output classifier (5th row), a unified detector with our learned unified label space (6th row), and a unified detector with the human expert label space (7th row). For reference, we show an "oracle" model that is trained on the training set of each test dataset on the bottom row. The columns refer to test datasets.

**Training loss.** The standard CascadeRCNN uses softmax cross-entropy as the classification loss (Ren et al., 2015). This is infeasible in our case. Softmax assumes one object has at most one class label, while OpenImages (Kuznetsova et al., 2020) requires predicting a label hierarchy (e.g., it requires predicting "vehicle" and "car" for all cars) and may have overlapping labels for the same object (e.g., both "toy" and "car" for a toy car). In our implementation, we use the sigmoid cross-entropy loss for all multi-dataset models and baselines, but use the best dataset-specific loss when training a baseline on just one dataset. For OpenImages, we use a hierarchy-aware sigmoid cross-entropy loss that sets parent classes as positives and ignores the losses over descendant classes. When training on multiple datasets, we also ignore labels outside the training dataset for each image. Appendix D reports ablation experiments on the hierarchy-aware and multi-dataset losses.

**Data sampling.** There is significant spread in the sizes of the training datasets (see Appendix A). In our experiments, we found that sampling images evenly from all training datasets works best. Following the current best practice (Peng et al., 2020), we use class-aware sampling (Shen et al., 2016) for OpenImages and Objects365 to tackle the long-tail class distribution. A controlled evaluation of the different sampling strategies can be found in Appendix E.

**Training details.** Our implementation is based on detectron2 (Wu et al., 2019) and we adopt most of the default hyperparameters for training. We use the standard object detection data augmentation, including random flip and random scaling of the short edge in the range $[640, 800]$. We use SGD with a base learning rate 0.01 and batch size 16 over 8 GPUs. We use ResNet50 (He et al., 2016) as the backbone in our controlled experiments unless specified otherwise.

For our label space evaluation experiments, we use a standard $2\times$ schedule (180k iterations with learning rate dropped at the 120k and 160k iterations) (Wu et al., 2019) to save computation. This results in 8 epochs on COCO, 0.5 epochs on OpenImages, and 1.6 epochs on Objects365. When comparing models trained on different datasets (Section 5.2 and Section 5.4), we use an $8\times$ schedule (720k iterations with learning rate dropped at the last 60k and 20k iterations) (Wu et al., 2019; Goyal et al., 2017) or until the model converges (for small datasets). A comparison of training schedules is provided in Appendix F.

## 5.2 ZERO-SHOT CROSS-DATASET EVALUATION

For our main evaluation, we train a detector on all four RVC training datasets and evaluate it on new datasets that were not seen during training. We train a strong disjoint label-space detector using a ResNeSt101 (Zhang et al., 2020) backbone with the same training procedure as described in Section 5.1. We obtain the predicted bounding boxes in the validation sets to run our label space optimization algorithm. We compare our unified detector to the disjoint label-space baseline fine-tuned with the same schedule, hyperparameters, and detection models. For reference, we also show the performance of detectors trained on the training set of each test dataset. This serves as an oracle "upper bound" that has seen the test domain and label space. Note that KITTI (RVC version) and

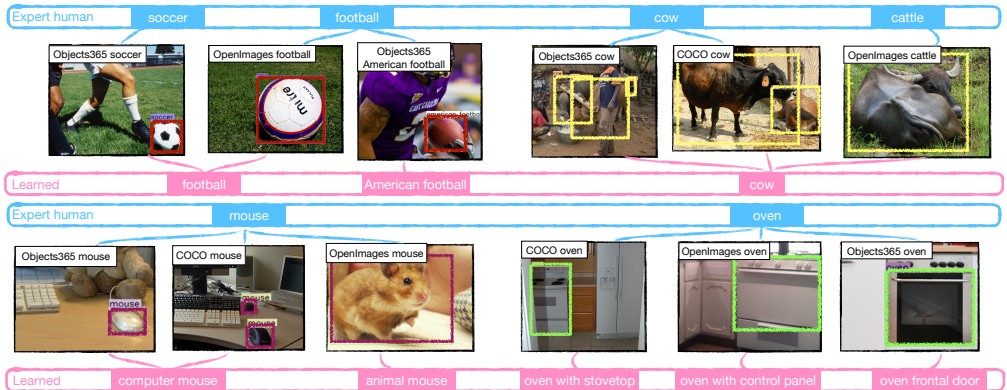

Figure 3: Example differences between an expert designed label space provided as part of the ECCV 2020 Robust Vision Challenge (top of each row, blue) and our learned label space (bottom of each row, pink). Best viewed on the screen.

WildDash are small and do not have a validation set. We thus directly evaluate on the training set and do not provide the oracle model.

Table 1 shows the results under the mEAP@0.5 (IoU threshold 0.5) metric. The COCO model already exhibits reasonable performance of some test datasets, such as Pascal VOC and CrowdHuman. However, its performance is less than satisfactory on datasets such as ScanNet, whose label space differs significantly from COCO. Training on the more diverse Objects365 dataset yields higher accuracy in the indoor domain, but loses ground on VOC and CrowdHuman, which are more similar to COCO. Training on all datasets, either with a disjoint label space for each dataset or with a unified label space yields generally good performance on all test datasets. The individual test datasets are closer to the span of all training datasets than to any individual dataset. Note that on Pascal VOC, our unified model outperforms the VOC oracle model without seeing VOC training images.

Our unified model consistently produces high-quality detections for all classes, while the quality of the outputs of the disjoint baseline depends on the respective dataset-specific label space. For example, a Mapillary person detector generalizes much more poorly than a COCO one. In addition, a disjoint baseline yields redundant detections per object.

## 5.3 EVALUATION OF THE UNIFIED LABEL SPACE

Next, we analyze our unified label space on COCO, Objects365, and OpenImages. We compare to a human expert label space officially provided as part of the ECCV Robust Vision Challenge, derived by the challenge organizers[1]. Our optimization gives 701 classes in the unified label space, which is more than the 659 classes in the human expert label space. This is because some semantically similar concepts between datasets have different visual expressions.

Figure 3 shows some examples of differences between our learned unified label space and a human expert label space. The learning algorithm can separate visually different categories with similar words ("American football" and "football"), and merge the same concept expressed in different words ("Cow" and "Cattle"). Interestingly, the learned label space splits COCO, Objects365 and OpenImages oven, even though they share exactly the same word. However, they are visually dissimilar: COCO ovens include the cooktop, OpenImages only the control panel, and Objects365 oven focuses just on the front door. This signal is only present in the visual data.

To quantitatively evaluate the label spaces in practice, we train a unified detector on each label space using the same training procedure (ResNet-50 and the $2\times$ schedule), and compare their detection performance. Table 2 shows the results. We additionally compare to a language-based baseline (Glove embedding). Specifically, we replace the cost measurement defined in Section 4.2 with the cosine distance between the Glove word embeddings (Pennington et al., 2014), and run the same integer linear program. The three label spaces obtained are similar in most classes, hence the overall mAP does not change much. However, our label space consistently outperforms the human expert,

---

[1]https://github.com/ozendelait/rvc_devkit/blob/master/objdet/obj_det_mapping.csv

|  | COCO | Objects365 | OpenImages | *mean* |
|---|---|---|---|---|
| Glove embedding | 41.6 | 20.3 | 62.3 | 41.4 |
| Learned, distortion | 41.5 | 20.7 | 62.6 | 41.6 |
| Learned, mAP (ours) | **42.0** | **20.9** | **62.8** | **41.9** |
| Expert human | 41.5 | 20.6 | 62.6 | 41.6 |

Table 2: Evaluation of unified label spaces. We measure mAP on the validation sets of the training datasets. We compare to a language-based baseline and a manual unification by a human expert.

|  | COCO | Objects365 | OpenImages | Mapillary | *mean* |
|---|---|---|---|---|---|
| Unified | 44.9 | 23.9 | 65.7 | 14.8 | 37.3 |
| Disjoint label-space oracle | 45.1 | 24.0 | 65.1 | 14.9 | 37.3 |
| Dataset-specific oracle | 42.5 | 24.9 | 65.7 | 15.5 | 37.2 |

Table 3: Detection mAP on the validation sets of the training datasets. We show the performance of our unified model, the disjoint label-space detector with the oracle head at test time, and dataset-specific models (the last row, where each column is from a different model).

with a healthy $0.3$ mAP margin in average. The relative improvement of our model over the expert is larger than the experts' improvement over the language-based baseline.

## 5.4 PERFORMANCE ON TRAINING DATASETS

Table 3 compares the unified detector, the disjoint label-space detector, and dataset-specific detectors on the four training domains. Our unified detector is competitive with the disjoint label-space detector without knowing the image domain at test time. On COCO, our unified detector outperforms the COCO-specific detector by $2.4$ mAP, likely because it benefits from $20\times$ more data provided by the other training datasets. On the other three datasets, the unified model matches the dataset-specific models within $1$ mAP. Our work shows that training on multiple datasets can increase the model's generality across domains without compromising accuracy within domains.

## 5.5 ECCV ROBUST VISION CHALLENGE

We submitted a model trained with the presented approach to the ECCV 2020 Robust Vision Challenge (RVC). We used a heavy ResNeSt200 backbone (Zhang et al., 2020) and followed the same training procedure as in Section 5.4 to train the model using an $8\times$ schedule. We used a unified label space of $682$ classes learned with the distortion loss. The training took $\sim16$ days on a server with 8 Quadro RTX 6000 GPUs. Table 4 summarizes the results of the challenge. Our model outperforms all other RVC entries on all datasets by a large margin. Notably, WiseDet_RVC used a stronger detector (Qiao et al., 2020), but without our learned label space or multi-dataset training setup (Section 5.1). The bottom rows of Table 4 show the state-of-the-art results reported on each individual dataset. On COCO, our result is comparable with DetectoRS (Qiao et al., 2020), which is by default $2.4$ mAP higher than our ResNeSt200 backbone ($50.9$ mAP) (Zhang et al., 2020). On OpenImages, our result matches the best single-model performance of the OpenImages 2019 Challenge winner, TSD (Song et al., 2020), with a comparable backbone (SENet154 (Hu et al., 2018) with deformable

|  | COCO | OpenImages | Mapillary | Objects365 |
|---|---|---|---|---|
| Ours | **52.9** | **60.6 / 56.8** | **25.3** | **33.7** |
| WiseDet_RVC | 40.0 | 56.1 / 53.3 | 22.5 | - |
| FRCNN_R50_GN_RVC | 34.0 | 21.4 / 19.9 | 8.1 | - |
| DetectoRS (Qiao et al., 2020) | **53.3** | - | - | - |
| TSD (Song et al., 2020) | - | 60.5 / - | - | - |
| CACascade RCNN (Gao et al., 2019) | - | - | - | 31.6 |

Table 4: Test set performance on RVC datasets: COCO test-challenge set, OpenImages challenge 2019 test sets (shown in public test set/ private test set), Mapillary test set, and Objects365 validation set. Top: results of RVC challenge participants. Bottom: the published state-of-the-art performance on each specific dataset (without model ensembles or test-time augmentation). Objects365 was initially part of the challenge but was removed in the final evaluation.

|  | COCO | CityScapes | Mapillary | VIPER | ScanNet | OpenImages | KITTI | WildDash |
|---|---|---|---|---|---|---|---|---|
| COCO | **35.6** | 19.6 | 3.2 | 8.5 | 5.2 | 7.2 | 15.7 | 8.4 |
| CityScapes | 0.0 | 21.5 | 0.8 | 2.3 | 0.0 | 0.0 | 13.0 | 2.4 |
| Mapillary | 0.6 | 11.7 | **10.6** | 9.0 | 1.2 | 0.0 | 13.4 | 5.4 |
| VIPER | 0.1 | 2.8 | 1.1 | **17.8** | 0.0 | 0.0 | 6.5 | 1.4 |
| ScanNet | 0.4 | 0.0 | 0.0 | 0.0 | **35.6** | 0.0 | 0.0 | 0.0 |
| OpenImages | 12.9 | 9.5 | 1.1 | 3.5 | 1.7 | **52.8** | 7.2 | 4.9 |
| Unified (ours) | 24.0 | **28.3** | 8.1 | 16.5 | 28.7 | 41.8 | **16.9** | **11.3** |

Table 5: Instance segmentation performance on six training datasets and two new datasets (KITTI and WildDash). We show mask mAP when the test dataset is included in training, and show mask mEAP when the testing on new datasets.

|  | COCO | CityS. | Mapillary | VIPER | ScanNet | OpenImages | KITTI | WildDash |
|---|---|---|---|---|---|---|---|---|
| Ours | 33.0 | 29.8 | 13.0 | 18.9 | 20.5 | 35.0 | 23.2 | 21.0 |
| seamseg_rvcsubset | - | 22.1 | - | - | - | - | - | 20.9 |
| EffPS_b1bs4_RVC | - | 21.3 | - | - | - | - | - | - |

Table 6: Leaderboard of RVC instance segmentation challenge. We show results on the test set for each datasets (test-challenge for COCO and private test set for OpenImages).

convolution (Zhu et al., 2019)). On Objects365, we outperform the 2019 challenge winner by 2 mAP.

## 6    INSTANCE SEGMENTATION

We further evaluate our label space learning algorithm and unified training framework on instance segmentation. We follow the ECCV Robust vision challenge set up to use 8 datasets: COCO, OpenImages, Mapillary, ScanNet, VIPER, CityScapes, WildDash and KITTI (the same as Table 7, except OpenImages segmentation set has 300 instead of 500 classes.). Again, we leave WildDash and KITTI as testing only as they are small and similar to CityScapes and Mapillary. We run our label space learning algorithm (Section. 4.1) on the remaining six datasets, resulting a unified label space of $358$ classes. We use CascadeRCNN (Cai & Vasconcelos, 2019) with a standard mask head as the detector, and train a $2\times$ schedule with ResNet50. The dataset-specific models are trained with $1\times$ or $2\times$ schedule depending on their size.

Table. 5 compares the unified detector (with instance segmentation) to dataset specific models. As expected, no single dataset-specific model performs well on all test domains. Our unified model performs consistently good on all training datasets. More importantly, it generalizes the best to the new test datasets (KITTI and WildDash) than any single dataset model. Table. 6 compares our method with others on the test sets of RVC instance segmentation challenge. We outperform other entries on all datasets that have a valid submission.

## 7    CONCLUSIONS

We presented an automated way to unify the label spaces of multiple datasets. This enables training a single detection model that works across the training domains and beyond. We showed that the resulting detector is robust in zero-shot cross-dataset testing. Our current implementation unifies matching concepts, but does not yet merge hierarchical concepts. Neither do we adapt our model to unseen data using unsupervised objectives. These are exciting avenues for future work.

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

| Dataset name | Domain | # Categories | # Images | Note |
|---|---|---|---|---|
| **Train & Validation** | | | | |
| COCO | Internet images | 80 | 118k | - |
| Objects365 | Internet images | 365 | 600k | Long-tail |
| OpenImages | Internet images | 500 | 1.8M | Federated/ Hierarchical label |
| Mapillary | Traffic | 38 | 18k | High-resolution |
| **Test** | | | | |
| ScanNet | Indoor | 20 | 25k | - |
| VIPER | Virtual | 10 | 13k | - |
| CityScapes | Traffic | 8 | 12k | High-resolution |
| WildDash | Traffic | 13 | 4k | Extreme driving scenes |
| KITTI | Traffic | 8 | 200 | - |
| Pascal VOC | Internet images | 20 | 16k | - |
| CrowdHuman | Internet images | 1 | 15k | Crowded |

Table 7: Summarize of datasets we used. Top: datasets we used in training, which are from ECCV 2020 RVC challenge; Bottom: datasets we used for zero-shot cross dataset testing. We list the features of each dataset in the last column.

## A  DATASET DETAILS

Table 7 lists the datasets we used in our object detection experiments. For ScanNet (Dai et al., 2017), as there is no standard train/ validation split, we use the first $80\%$ scenes (sorted by scene ID) as training and the last $20\%$ scene as validation. For KITTI (Geiger et al., 2012), we used the RVC challenge that has instance-segmentation version, which contains 200 images. For WildDash (Zendel et al., 2018), we use the public version for evaluation, and report standard mAP performance. We don't consider the negative label metric in the official website. For CrowdHuman (Shao et al., 2018), we use the visible bounding box annotation, and report the standard mAP instead of the missing rate as the official metric.

## B  LABEL SPACE EXPANSION ALGORITHM

While we tend to keep the training domains and label space large and comprehensive, it is inevitable in practice that more fine-grained labels or specific testing domains are needed. Given a learned a unified label space on an existing set of training datasets, we propose a label space expansion algorithm to allow adding more datasets and labels after the unified detector is trained.

Similar to our unified label space learning algorithm, we run the unified detector on the new training data. We evaluate the mAP between each class in the new dataset annotation and each class in the unified label space. We merge the new class into the existing class that gives the lowest merge cost (Section. 4.2), if the cost is lower than a threshold (mAP change $< 5$ mAP in our implementation). Otherwise, we append the new class to the unified label space as a single class.

## C  mEAP DETAILS

For each test label, we first find training labels with closet Glove embedding distance. The same word has the Glove distance 0. For a joint class in a unified label space, it may has different names from different datasets. We set the embedding distance as the minimal embedding distance among its composing classes. In the case that there is no training label close to a test label (i.e., minimal Glove distance $> \theta = 0.6$) due to limited label space, we collect all labels that contain or being contained the test class as the corresponding labels.

There can be more than one training label corresponds to a test label, when the class in the same name from two different training datasets are not merged or there is finer label granularity in the training label space. In this case, a human user can pick any training class within the corresponding set for the test class during real-world application. To mimic this in evaluation, we define a new

|  | COCO | Objects365 | OpenImages | *mean* |
|---|---|---|---|---|
| sigmoid cross entropy loss | 41.5 | 20.1 | 58.4 | 40.0 |
| +multi-dataset loss | 42.0 | 20.8 | 61.4 | 41.4 |
| +hierarhical-aware loss (ours) | 42.0 | 20.9 | 62.8 | 41.9 |

Table 8: Ablation experiments on training losses. We start with a sigmoid cross entropy loss, and add our multi-dataset loss and hierarchical-aware loss (for OpenImages only) one by one. Experiments are conducted on ResNet-50 with $2\times$ schedule.

|  | COCO | Objects365 | OpenImages | *mean* |
|---|---|---|---|---|
| w.o. class-aware sampling | 41.4 | 16.8 | 49.4 | 35.9 |
| w.o. evenly sampling between datasets | 35.3 | 18.5 | 65.1 | 39.6 |
| Ours | 42.0 | 20.9 | 62.8 | 41.9 |

Table 9: Ablation experiments on data sampling. We compare sampling by the original size of each dataset (first row) to evenly sampling across datasets, and validate the effectiveness of class-aware sampling for Objects365 and OpenImages (second row).

metric, expected AP (EAP), that calculates the class-specific AP as the average (expectation if a user chooses the corresponding class randomly) AP of all the corresponding training class for a test class. The overall expected mAP (mEAP) on a dataset is the average E-AP overall classes as the conventional mAP.

## D   Ablation studies on training loss

**Multi-dataset loss** When training a unified label space, the annotations in each dataset becomes incomplete. For example, there is no "fish" class in COCO, but a COCO image may contain a fish as background. As the unified label space has a fish class (from OpenImages), a naive softmax cross-entropy loss will mistakenly train the COCO fish bounding box as a negative sample for the unified fish classifier. To facilitate this issue, we use a modified sigmoid cross-entropy loss to ignore the output from out-of-source-dataset classes. I.e., when training COCO images, only the 80 COCO classes are applied a loss.

Table 8 ablates different losses for training on multi-dataset. We start with a sigmoid cross-entropy loss for classification. This is $\sim$ 0.5 mAP worse than softmax cross-entropy on single dataset training (results not shown). The advantage of sigmoid cross-entropy is it breaks the inter-class dependency and makes ignoring classes easier. We add the multi-dataset loss that ignores labels outside the source dataset. This gives in average 1.4 mAP improvements to the three datasets. Further, to facilitate the hierarchical-aware evaluation on OpenImages, we use a hierarchy-aware sigmoid cross-entropy for OpenImages samples that sets parent classes as positives, and ignores child classes. This gives 1.4 mAP improvement on OpenImages, and is compatible with other datasets.

## E   Ablation studies on data sampling

We apply class-aware sampling (Shen et al., 2016) to Objects365 and OpenImages, and sample COCO images uniformly. As is shown in Table. 9, removing class-aware sampling drops 6 mAP in

|  | $2\times$ | | | $6\times$ | | | $8\times$ | | |
|---|---|---|---|---|---|---|---|---|---|
|  | COCO | Objects365 | Oimg. | COCO | Objects365 | Oimg. | COCO | Objects365 | Oimg. |
| Unified | 42.0 | 20.9 | 62.8 | 44.6 | 23.3 | 64.5 | 45.4 | 24.4 | 66.0 |
| COCO | 41.5 | - | - | 42.5 | - | - | 42.5 | - | - |
| Objects365 | - | 23.8 | - | - | 25.0 | - | - | 24.9 | - |
| OpenImages | - | - | 64.6 | - | - | 65.4 | - | - | 65.7 |

Table 10: Ablation studies on trining schedule. We train the unified detector on and each dataset-specific models for different training schedules and show mAP on each training datasets.

| notation | definition | dimension | range |
|---|---|---|---|
| $N$ | number of datasets | scalar | integer |
| $i$ | dataset index | scalar | 1-$N$ |
| $\hat{L}_i$ | set of labels of dataset i | $|\hat{L}_i|$ | 1 - $|\hat{L}_i|$ |
| $\hat{L}$ | set of labels in the disjoint label space | $|\hat{L}| = \sum_i |\hat{L}_i|$ | 1 - $|\hat{L}|$ |
| $L$ | set of labels in the unified label space | $|L|$ | 1 - $|L|$ |
| $K$ | number of objects in dataset i | scalar | integer |
| $\hat{\mathbf{l}}_i$ | labels of all objects in dataset i | $K$ | $[1, |\hat{L}_i|]$ |
| $\mathbf{b}_i$ | ground truth bounding boxes of all objects in dataset i | $K \times 4$ | $\mathbb{R}$ |
| $\tilde{\mathbf{l}}_i^{(j)}$ | predicted labels in head j of all objects in dataset i | $K$ | $[1, |\hat{L}_j|]$ |
| $\mathbf{l}_i$ | labels of all objects in dataset i in the unified label space | $K$ | $[1, L]$ |
| $\mathcal{T}_i$ | transform function from label space i to the unified label space | $|\hat{L}_i| \times |L|$ | $\{0, 1\}$ |
| $\mathbf{t}$ | potential merges (a set of class indexes) | $< N$ | $[1, |\hat{L}_i|]$ |

Table 11: Table of notations used in the algorithms. For each notation, we also list their dimension and the value range.

average for the three datasets. The OpenImage training set is $3\times$ as Objects365, and $15\times$ as COCO. An alternative sampling strategy is to sample by this ratio. The outcome is it trades-off the small dataset performance for large dataset performance, and gives an overall lower average mAP.

## F    ANALYSIS ON TRAINING SCHEDULES

Table. 10 shows how the performance evolves when training goes longer. In a $2\times$ schedule (180k iterations), all dataset-specific models converge within a $1.2$ mAP gap comparing to the $8\times$ schedule. The unified model is not fully converged, as each dataset is only trained for a $\frac{2}{3}\times$ schedule. From a $2\times$ schedule to a $6\times$ schedule, the gaps between the unified model and the dataset-specific models are narrowed. In the final $8\times$ schedule, the unified model surpasses the single dataset-model on COCO and OpenImages, and closely matches the Objects365 model.

## G    ALGORITHM DIAGRAMS

Algorithm 1 outlines the workflow for training our unified object detector. Algorithm 2 summarizes the algorithm we used for learning a unified label space (Section 4.1). We list the notation definitions of both algorithm in Table. 11.

---

**Algorithm 1:** Training a unified detector

---

**Input** : $\{\mathbf{x_i}, \mathbf{b_i}, \hat{\mathbf{l}}_\mathbf{i}\}_{\mathbf{i=1}}^{\mathbf{N}}$: labeled training datasets
**Output:** $L$: a unified label space of all datasets
    $\mathcal{M}$: an object detector on the unified label space $L$

1 $\mathcal{M}_0 \leftarrow$ train_detector($\{\mathbf{x_i}, \mathbf{b_i}, \hat{\mathbf{l}}_\mathbf{i}\}_{\mathbf{i=1}}^{\mathbf{N}}$) // train an initial object detector with a disjoint label space $\hat{L}$ on all datasets

2 **for** $i \leftarrow 1\ to\ N$ **do**

3 $\quad \{\tilde{\mathbf{b}}_i^{(j)}, \tilde{\mathbf{l}}_i^{(j)}\}_{j=1}^N \leftarrow \mathcal{M}(\mathbf{x}_i)$ // run the disjoint label space detector on each dataset, and get boudning boxes and label predictions from each detection head.

4 **end**

5 $L, \mathcal{T} \leftarrow$ learning_unified_label_space($\{\mathbf{b}_i, \hat{\mathbf{l}}_i\}_{i=1}^N, \{\{\tilde{\mathbf{b}}_i^{(j)}, \tilde{\mathbf{l}}_i^{(j)}\}_{j=1}^N\}_{i=1}^N$) // Algorithm 2

6 **for** $i \leftarrow 1\ to\ N$ **do**

7 $\quad \mathbf{l}_i \leftarrow \mathcal{T}_i(\hat{\mathbf{l}}_i)$ // transform labels form its original label space to the unified label space

8 **end**

9 $\mathcal{M} \leftarrow$ train_detector($\{\mathbf{x_i}, \mathbf{l_i}, \mathbf{b_i}\}_{\mathbf{i=1}}^{\mathbf{N}}$) // train an unified detector with labels in the unified label space

10 **Return**: $L, \mathcal{M}$

---

**Algorithm 2:** Learning a unified label space

---

**Input** : $\{\mathbf{b}_i, \hat{\mathbf{l}}_i\}_{i=1}^N$: ground truth bounding boxes and labels for each training dataset
    $\{\{\tilde{\mathbf{b}}_i^{(j)}, \tilde{\mathbf{l}}_i^{(j)}\}_{j=1}^N\}_{i=1}^N$: predicted bounding boxes with predicted classes in all datasets
for each training dataset
**Output:** L: unified label space
    $\mathcal{T}$: the transformation from each individual label space to the unified label space

1 // Compute potential merges and merge cost

2 $\hat{L} = \bigcup_i \hat{L}_i$ // Short-hand used to simplify notation

3 $\mathbb{T}_1 \leftarrow \{(l)|l \in \hat{L}\}$ // Set of single labels

4 Compute $c_t$ for all single labels $\boldsymbol{t} \in \mathbb{T}$. // 0 for most metrics

5 **for** $n = 2 \ldots N$ **do**

6 $\quad \mathbb{T}_n \leftarrow \{\}$

7 $\quad$ **for** $\boldsymbol{t} \in \mathbb{T}_{n-1}$ **do**

8 $\quad\quad$ **for** $l \in \hat{L}$ **do**

9 $\quad\quad\quad$ **if** $l$ *and all labels in* $\boldsymbol{t}$ *are from different datasets* **then**

10 $\quad\quad\quad\quad$ compute $c_{\boldsymbol{t}\cup\{l\}}$.

11 $\quad\quad\quad\quad$ **if** $\frac{c_{\boldsymbol{t}\cup\{l\}}}{n-1} \leq \tau$ **then**

12 $\quad\quad\quad\quad\quad$ Add $\boldsymbol{t} \cup \{l\}$ to $\mathbb{T}_n$.

13 $\quad\quad\quad\quad$ **end**

14 $\quad\quad\quad$ **end**

15 $\quad\quad$ **end**

16 $\quad$ **end**

17 **end**

18 $\mathbb{T} \leftarrow \bigcup_{n=1}^N \mathbb{T}_n$

19 // Solve the ILP.

20 $\boldsymbol{x} \leftarrow$ ILP_solver($c, \mathbb{T}, \lambda$) // Solve equation (4).

21 Compute $L, \mathcal{T}$ from $\boldsymbol{x}$

22 **Return**: $L, \mathcal{T}$

---

