# OpenReview forum: "Learning a unified label space"
_ICLR.cc/2021/Conference — Reject_

### Official Review · AnonReviewer1 · 2020-10-28
**For robust training a unified label space is essential**

**Rating:** 7
**Confidence:** 4

**Review:**

**Paper Summary and strengths**
In this paper robust object detection is studied, that is a single network is trained over multiple datasets and evaluated on datasets not seen during training [as for example promoted by the robust vision challenge, or the MSEG datasets]. One common problem for these approaches is how to deal with the different label spaces of each datasets, naive approaches would define an union over all label spaces, which has the undesirable effect that a CAR in one dataset is a different CAR in another dataset. In this paper a method is proposed to map this union label space into data specific subspaces by grouping similar labels based on their object classifier scores. In this paper this is solved by  a linear integer program which finds the optimal number of labels in the unified label space and their mappings and weights. Moreover this approach shows promising results over the disjoint/union label space.

**Weaknesses**
1. Missing baseline: The main results in Table 1 do not compare to the simple 'best effort name matching' as provided by the Robust Vision Challenge dev kit. This would be interesting to see the difference in performance between 'semantic matches' (name matching) and 'visual matches' from the provided algorithm. Also the influence of lambda (on the compactness of the unified label space) should be studied, by a very large lambda value, all labels will be mapped to a single label.

2. At first sight the method is conceptually extremely straightforward, define a joint label space, with a permutation to the dataset specific label spaces (with some constraints to make it direct mappings) and re-train a final linear layer on top of pretrained detector. However, the description is difficult to follow with the different symbols, super and sub scripts, and slightly different terminology used throughout the paper [union, disjoint label spaces]. I think this should be carefully considered since this is the most important section of the paper. Maybe psuedo code or a more illustrative description can improve the presentation of the method. [why is at most 1 label per dataset merged? That only holds for a specific label in the unified space]

3. Use the unified label space for training an end-to-end detector. The proposed method is only feasible since a pre-trained detector is used, cf Section 4. It would be interesting to see if an end-to-end trained detector on the found unified label space is able to boost the performance even further.

4. Not really a weakness, but rather a possible extension to consider: In the current method only the visual appearances of the labels are used for constructing the unified label space. It would be interesting to also use the semantics of the provided labels [for the current method it doesnt matter whether a label is called CAR or BMNASDJHASD].

**Minor**
- Page 3, Eq 1: N is undefined.
- Page 7, Figure 3: 'Expert human' is largely overrated for the automatic name matching process used by the Robust Vision Challenge. [From their dev kit: We do provide a "best-effort" mapping, which can be a good starting point. This mapping will contain overlapping classes and some dataset entries might miss relevant labels.] It would also be good to show some 'failure' or 'unexpected' merged classes from the different datasets.
- Please include the Instance Segmentation results in the main body of the paper.

**Conclusion**
This paper provides a method for learning a unified labels space when training a task over multiple datasets. This is timely and highly relevant with the current focus on robust methods in various application of computer vision [Robust Vision Challenge, MSEG, VTAB, ...]. The paper is somehow difficult to follow, but that should be solvable within the rebuttal phase.

**Post-Rebuttal**
After reading the rebuttal, the updated manuscript and the other reviews, I became *less* convinced about this paper. While I remain positive about the conceptual idea of learning a unified label space. The authors did not successfully convince me in improving the understanding of the method. The pseudo-code alone does not make the  algorithm better to understand. It seems that the space T is insanely big (100x100x100 when three datasets of 100 labels are used). Also, while the authors do add a requested baseline, it is hardly compared (afaik only in the sentence that the expert human obtains 659 labels, while the learned space contains 701). Also note that the 'human expert' is a *"best-effort" mapping, which can be a good starting point.* according to the RVC dev kit. So, this might be an overclaim.

Finally, please carefully proof read the paper, main typos remain.

---

> ### Author Response · Authors · 2020-11-24
> **Reply to AnonReviewer1**
>
> Thank you for the constructive feedback! We have revised the paper to clarify all questions accordingly.
>
> Q: What is the performance of the human-expert baseline in the setting of Table. 1?
>
> A: Thank you for pointing out the missing baseline! We have added the human-expert results in Table.1 of the updated paper. This expert human baseline performs roughly as well in terms of generalization to new domains, slightly underperforms our model in the training-domain (Table. 2).
>
> Q: The formulation of Section. 4 is a bit complex. A pseudo-code or a more illustrative description can improve the presentation of the method.
>
> A: Thank you for the nice suggestion! We have added an algorithm diagram in supplement G  of the updated paper.
>
> Q: Why is at most 1 label per dataset merged?
>
> A: By “Each common label maps to at most one dataset-specific label per dataset, and each dataset-specific label maps to exactly one common label.”, we mean we do not map two different classes in one dataset to the same unified label. This follows the fact that classes within each dataset are not duplicates of each other. Duplicates only occur across datasets.
> This does change when considering hierarchies, which we have not tackled in this project.
>
>
> Q: Can we train a detector with the learned label space in an end-to-end way?
>
> A: Yes, actually our detector is trained end-to-end. In Section. 4, we formulate our unified-label space learning algorithm as finetuning the last layer, because anything else would be intractable. However, once we have a learned unified-label space, we train the network from ImageNet initialization with the learned unified label space. And yes, we observe training the network end-to-end with the learned label space performs better than merging the last layer weights offline. We updated the writing in the paper to reflect this.
>
>
> Q: Can we learn a label space with combined visual and semantic features?
>
> A: In theory yes. We can set the optimization objective as a weighted combination of the current visual term and a word-embedding distance term, but haven’t experimented with this yet.
>
> Q: It would also be good to show some 'failure' or 'unexpected' merged classes from different datasets.
>
> A: Our learned label space merges classes conservatively. It resulted in more classes in the unified space than the human merge, and we didn’t find significant failure cases.
> Interesting differences. One counter-intuitive merge we note is we merged “infant bed” in OpenImages with COCO bed and Objects365 bed, while leaving OpenImages bed as an individual class. This is because our current formulation does not hierarchically relate concepts across datasets, and OpenImages bed also includes a “studio couch”. We leave joint learning the hierarchical label relationship across datasets as exciting future work.
>
> Q: Please include the instance segmentation results in the main paper.
>
> A: Thank you for the suggestion. The updated paper has them in the main paper.

---

### Official Review · AnonReviewer3 · 2020-10-28
**The paper aims to train a unified detector that works well across datasets by constructing a common label space that is share among these datasets.**

**Rating:** 4
**Confidence:** 4

**Review:**


The general idea of the paper is clearly interesting: Training a unified detector across datasets by also optimizing for a common label space. The formulation to address the problem discussed in section 4 seems to make sense.

The experiments, however where quite confusing and inconclusive  in my view.

- The "main experiments" of the paper are so called "zero-shot" results, while the paper never defines what this is referring to. The zero-shot setting that I know of (as e.g. defined on the Animals with Attributes dataset) does not seem to apply here as no "novel class labels" can be handled by the algorithm. Therefore it is not clear to me what zero-shot means in this paper.

- Additionally, the main results are shown in table 1, while it is not clear what the columns and lines refer to. This might be linked to the issue that zero-shot is not defined, but as a reader I cannot read the table without a proper description of the table. As a reader I am left with guessing

- The authors introduce a novel metric (mEAP) and it is unclear why this should be a better or more suitable metric. I see no reason why standard metrics could not be used when evaluation is done on a per dataset level. If I am not mistaken this is what is done in tab 1 - but again too little information is given to know for sure. The definition of mEAP discusses vaguely some mapping using word embeddings - but this leaves lots of room for speculation and the "detailed" description in append C is very ad hoc with arbitrary thresholds. As a reader I have no real understanding if this new measure is sensible as no attempt is made to convince the reader that this is a good measure.

- Other results are reported in standard measures in table 2, 3 and show very small differences between different methods thus showing no real advantage of the proposed method.

- linked to the first point: please make clear if the experiments are in a "generalized zero-shot setting" that is that all classes are present during testing or not and if the classifier output can be all classes or only the classes of the tested dataset, if the accuracy is with respect to all classes of the classifier or only with respect to the classes of the dataset, if training is done in a "transductive" setting or not (I assume not?), etc

So while overall the papers aims for an interesting goal, the reported results are either not conclusive or obscured (using an unclear setting and a novel metric that is unknown to the readers) where the conclusions are unclear as well.

A minor note: The paper promises to make code available upon acceptance. However, given the fact that generation of the respective models will be beyond reach for many researchers, the authors should share also share the trained models and not only the code.

Additional comments after the rebuttal phase:

On the positive side
- the authors are clarifying a few things in the rebuttal and also in the paper such as table 1 - for that table results are clearer now by updating the table and caption.

On the negative side, however, I am less convinced after the rebuttal than it looked to be prior to the rebuttal. Let me give some of the most important things

- claiming that the results in table 2 and 3 are statically significant is hard to believe. The authors claim that they are but it is not clear what the mean by "standard deviation in mAP for each of the datasets... is within .1mAP" in their rebuttal - what is varied to get this standard deviation? Learning a detector e.g. with different random seeds will result in much larger differences than 0.1mAP - thus claiming this it is statically significant is actually not scientifically credible to me

- while I understand the arguments the authors make while standard evaluation metrics are potentially not ideal or comprehensive, I am still quite strongly unconvinced about the novel metric mEAP - that seems very specific for the setting used here and does not lent itself for easy understanding and is also dependent on some threshold that is not clear how to choose. In that sense all the experiments using that particular measure are still not convincing to me. Additionally, the authors do not make a real attempt to make this measure more accessible in any way. It is mostly stated that mAP does not work. Even though, as the authors point out, for most labels there is a "joint" label across datasets which allows to evaluate that directly at least for those joint labels (and these are the majority of classes apparently). Also in the rebuttal the authors simply defend their metric rather than to acknowledge that this is not particularly useful. As said - rather unconvincing and I am sticking with that. The rebuttal is making me even less convinced about that metric as no attempt is made to show that the metric is sensible and fair.

- I strongly recommend to NOT use the term zero-shot. It is not only confusing as mentioned before but also does not really apply for most labels (the authors mention themselves that for most labels there is a corresponding label in the other dataset) - thus is more of a domain-adaptation or label-adaptation problem than really a zero-shot setting. The authors defend the usage of the term zero-shot which I do not find praticularly unconvincing.


minor
- the so called "zero-shot cross-dataset generalization" setting is not properly defined. It is mentioned at the beginning of sec 5 without being properly introduced what really is meant.

- typo first line sec 4.1: detectpr -> detector

I really would have liked to see a strong rebuttal given the good results for the ECCV challenge and the importance of the problem. However, the rebuttal nearly caused me lowering the score. So overall the rebuttal has made me less convinced about the paper than before. Sorry to say.

---

> ### Author Response · Authors · 2020-11-24
> **Reply to AnonReviewer3 (Part 1/2)**
>
> Thank you for the positive feedback ("The general idea of the paper is clearly interesting").
> We clarified the concerns raised in the review in an updated version of the paper and will briefly discuss the comments below:
>
> Q1: What is “zero-shot” in the main evaluation referring to? They are different from the zero-shot classification in the AwA dataset. How does the proposed algorithm handle novel class labels?
>
> A: In its purest definition zero-shot simply refers to evaluating on domain X without training on it (see [1, 2]). The reviewer is right that attributes are a common way to tackle zero-shot classification or detection because they allow for a transfer of semantics. However, it is not the only way (see [3, 4] for other examples). In our work, we directly use the name of classes themselves to do the transfer. If our training datasets contain a sufficiently large vocabulary, zero-shot generation is simply a mapping from training concepts to testing ones.
>
> As discussed in the related work section, this work is complementary to existing zero-shot approaches: we aim to build a sufficiently large, ideally all test categories can be perfectly matched to a training category.
>
> [1]  Recent advances in zero-shot recognition: Toward data-efficient understanding of visual content, Fu et al. Signal Processing Magazine, 2018
>
> [2] MSeg: A composite dataset for multi-domain semantic segmentation, John et al. CVPR, 2020
>
> [3] Zero-shot learning by convex combination of semantic embeddings, Norouzi et al. ICLR, 2014
>
> [4] Zero-shot object detection, Bansal et al. ECCV, 2018
>
>
> Q2: Why do we need a new evaluation metric mEAP? Why can’t we use the standard mAP? The description in Appendix C is ad hoc and requires a hyper-parameter.
>
> A: The key difference between our cross-dataset evaluation and standard mAP evaluation is we don’t have the train-test label correspondence. Unless a human expert sits down and designs mappings from each learned taxonomy to the test set, the mapping is not unique. For most test classes, the correspondence is straightforward, when we have the exact test label in our training label space. In this case we can apply the standard AP, and it equals the evaluation result of EAP. However, for some test classes we don’t have the exact match in our training label space. For example, when we evaluate the detection performance of class “rider”, there is no exact “rider” class in the training label space. We can not directly apply the standard AP metric for “rider”. There are labels “rider--bicyclist” and “rider--motorcyclist” in our training label space (from the Mapillary dataset). Our strategy is to compute the AP of both “rider--bicyclist” and “rider--motorcyclist”, and provide their average as the final metric (expected-AP).
>
> The mEAP metric mimics a common application scenario: a user wants an object detector for a specific class X, and a human expert may hand over any related classes to the user. The mEAP simply marginalizes the stochasticity in the mapping from training to test taxonomy out. We find this preferable to a human alternative that is both subjective and not repeatable.
>
> As for the thresholds, they are unfortunately necessary due to the scaling of the embedding space of word embeddings. However, the mEAP is not very sensitive to them. They standardize the comparison, we would rather direct the reader to our code for mEAP than a description, as is customary with most benchmarks (see COCO mAP…).
>
>
> Q3: What are the columns and rows in Table. 1 referring to?
>
> A: The top 4 rows refer to detection models trained on a single dataset, the next rows are trained on all datasets, either by producing disjoint output or a unified label space (our method). The last row is for reference only and lists a dataset-specific baseline, evaluated just on that dataset. The columns refer to test domains. We updated the table caption to reflect this.
>
>
> Q4: The improvements reported in Table. 2 and Table. 3 are not significant.
>
> A: The reviewer is absolutely right for Table 3 (mAP on training datasets). That was the point we were trying to make. A unified model works as well as a disjoint or dataset-specific one on the training domain (none is significantly better than the other). This is a good sign, training a unified model is not a trade-off between training-domain performance and generalization. The main point of our paper is to show generalization to new domains arises when training on multiple training domains using a properly unified detector. This generalization happens less when training a disjoint label space, or on a single domain (see Table 1).
>
> The results in Table 2 are statistically significant. The standard deviation in mAP for each of the datasets and detection models is within 0.1 mAP. This makes an improvement of 0.3 statistically significant with a p-value of 2.5% and an improvement of 0.5 (on COCO) is significant with a p-value of 1%.

---

> > ### Author Response · Authors · 2020-11-24
> > **Reply to AnonReviewer3 (Part 2/2)**
> >
> > Q5: In the zero-shot detection setting, what classes does the detector output? What classes are evaluated? Is training done in a transductive way?
> >
> > A: The detector always outputs all classes in its label space, but not all of them are evaluated. When evaluating on a specific test domain, only the classes in the test domain are evaluated. The final metric (mEAP or mAP) is always an average of all classes in the test dataset. The training is not done in a transductive way. In our updated paper, we included the pseudo code for our training pipeline in Appendix G. We hope these illustrations can clarify the confusions in training.
> >
> >
> > Q6: The authors should also share their models, not just code.
> >
> > A: Thanks for the reminder! We will release our models together with the code, and updated the sentence in the paper accordingly.

---

### Official Review · AnonReviewer2 · 2020-10-28
**The paper proposes to learn object detection model, while training on different datasets with different, potentially overlapping, label spaces. While previous methods do the label space mapping, from each dataset specific label space to the common universal label space, manually, this paper proposes to learn such mapping automatically.**

**Rating:** 7
**Confidence:** 5

**Review:**


Summary
The paper proposes to learn object detection model, while training on different datasets with different, potentially overlapping, label spaces. While previous methods do the label space mapping, from each dataset specific label space to the common universal label space, manually, this paper proposes to learn such mapping automatically. The proposed models work as well as dataset-specific models on resp. training datasets and generalize far better.


Positives
- The paper is well written and well organized, it is very easy to appreciate what is being done
- The problem formulation of mapping each dataset label space to a joint label space using Boolean linear transforms, and integer programming formulation is novel and interesting
- Results are given on challenging datasets which were part of the ECCV Robust Vision Challenge (RVC), and the experiments and the performances of the proposed method are convincing. The proposed method was one of the top performing method in ECCV2020 RVC

Negatives
- The shortcomings of the method are:
* the method does not train the detectors end to end; it trains a final projection layer, which is put on top of the penultimate layer features of individually trained object detectors
* (if I understand correctly) the training is done only with annotated objects in the different datasets, i.e. if there is a face object in the image of a dataset which doesn't have face in the label space (e.g. COCO does not have face label, but faces appear in the images of the dataset), that will not be used for training face part of the detector. In this respect, what happens with background boxes? During the label space merging background (boxes which do not belong to any object in any dataset) is completely ignored?
- For evaluation on new datasets, as the authors noted, annotation over all the labels in the unified label space would be required. Zhao et al. provide such a test set, for a different collection of dataset, which might be useful in the future to try

These points should be discussed more in the paper.

Overall the paper is well written and has a novel formulation and solution to the problem of label space merging. It also evaluates the method convincingly on challenging public benchmarks.

---

> ### Author Response · Authors · 2020-11-24
> **Reply to AnonReviewer2**
>
> Thank you for taking time to review our paper and the positive comments! We have revised the paper to clarify the confusion. We respond to the specific questions below.
>
> Q: The network is not trained end-to-end. It just finetunes the last classification layer.
>
> A: Sorry for the confusion. We train our final network end-to-end. The optimization in Section 4 only considers the last layer, because anything else would be intractable. However, once the unified-label space is learned, we retrain the network from ImageNet initialization with the learned unified label space. We have highlighted this training scheme at the beginning of the technical section in the updated paper.
>
> Q: What happens to objects that are annotated in one dataset, but not in all datasets? Will it be mistakenly trained as background?
>
> A: This is a great question! Our approach trains two detectors: an initial detector in a disjoint label space (i.e., each training dataset has its own classification layer), and a joint detector.
> When training the disjoint detector, each dataset has its own label space and output. Yes, it is true that for dataset A class i may be an object, and for dataset B class i may be a background. However, the separate outputs resolve this issue. For example, when training an image from dataset B that contains an un-annotated class i, it will not contribute to any negative loss to output head A.
>
> For the final detector, we used a multi-dataset loss to handle this issue (described in Section 5.1 “training loss” and ablated in Appendix D). We only apply a training loss to the output logits that exist within that dataset and ignore all others. If class i in dataset B was a background, it would neither receive a positive or negative loss on images from dataset B, but it would still receive a positive loss from datasets that contain class i (e.g. dataset A). We have added these discussions to Appendix D in the updated paper in company with the multi-dataset loss.

---

### Official Review · AnonReviewer4 · 2020-10-29
**Nice approach, very relevant for applications, lacking some technical details and experiments**

**Rating:** 6
**Confidence:** 4

**Review:**

The main idea of the proposed work is to learn a universal label space for a given task (say object detection) and a set of different datasets with  semantically overlapping labels. The only supervision required by the approach is constituted by the single dataset label spaces and respective annotations. Each dataset label space may have partial o complete overlaps (e.g. rider mapping to cyclist and motorcyclist ). The approach exploits a pre-trained detector on the trivial label space given by the union of all label spaces as a starting point. An optimization problem jointly minimizing some loss taking into account the task error with a penalization on the cardinality of the label set.

Strengths

- Unsupervised approach not requiring to hand design hierarchy or label space correspondence
- Results are above state-of-the-art (challenge winner)
- Elegant formulation via constrained optimization

Weaknesses

- missing clear specification of loss in the optimization problem. The statement: "The loss function in our constrained objective 1 is quite general and captures a wide range of commonly used losses. We highlight two: an unsupervised objective based on the distortion of the output compared to the pretrained model, and mean Average Precision (mAP)." is confusing, is that THE loss you optimize in problem (1), is there something missing? a full clear specification would enhance the reproducibility of this work.
- lack of simpler baselines in evaluation.  The approach can be seen as a method to merge dataset annotations; a trivial solution to this is to cluster samples according to some feature space and clustering algorithm and then retrain. This approach has the same requirements in terms of supervision and the final result (e.g. cardinality of final label space) can be controlled via the clustering algorithm. Why this has not been evaluated as a simpler baseline? This would look a lot like [a]
- is the optimization only performing fine-tuning on the last layer? wouldn't be better to perform an actual fine-tuning of the whole detector according to the new found label spaces?
- no comparison with handlabelling counterparts. In the intro two works Zhao 2020 and Lambert 2020 are referred as manual counterparts of the proposed method. While it is not expected that for the presented approach to perform better than this kind of algorithms it is surprising that this kind of comparison is completely avoided.

The problem at hand is very interesting, especially for industrial applications and the formulation is elegant and leads to superior results with respect to other approaches. The lack of some experiments and the not so clear specification of the loss (especially from a reproducibility point of view) are the main concerns I have and why the current paper rank is marginally above the acceptance threshold.


References

[a] Deep Clustering for Unsupervised Learning of Visual Features, 2018

---

> ### Author Response · Authors · 2020-11-24
> **Reply to AnonReviewer4**
>
> Thank you for the constructive feedback! We have revised the paper to clarify all questions accordingly.
>
> Q: What is the exact formulation of the optimization objective?
>
> A: Thank you for pointing out the confusion. Equation (4) is our optimization objective function. We discussed two forms of the cost $c_{\bf t}$, as Equation (5) and Equation (6). Equation (6) was used for the bulk of the results in the paper. Equation (5) was used for the initial challenge results. In practice, we found Eq (6) to work slightly better if labeled validation data is available in the source domains.
>
> In this rebuttal, we further included the pseudo-code for our unified label space learning algorithm in Appendix G in the updated paper. We hope these illustrations can clarify the reproducibility concern. We also ensure to publish all code and models for full reproducibility.
>
>
> Q: Lack of baseline comparison to clustering algorithms.
>
> A: We felt that the difference between clustering and our algorithm is quite small. For example, if we replace the exclusivity constraint in Eq (1) with an upper bound on the number of merged classes Eq (1) is the K-means objective. The issue with the original k-means solver however is that it can and will merge classes from the same dataset, and violates the constraints in Eq (1). This would make it hard to fairly compare to baselines such as manually merged datasets, or the RVC challenge.
>
>
> Q: Is the optimization performed only for the last layer?
>
> A: Thank you for pointing out the confusion. Actually, we trained our network end-to-end. In Section. 4, we formulate our unified-label space learning algorithm as fine-tuning the last layer, because any other formulation would require considering all potential merges jointly and make the optimization intractable. However, once we have a learned unified-label space, we retrain the network from ImageNet initialization with the learned unified label space. And yes, we observe training the network end-to-end with the learned label space performs better than merging the last layer weights offline. We have highlighted this training scheme at the beginning of the technical section in the updated paper.
>
> Q: Comparison to human expert label space.
>
> A: In Section 5.3, Figure 3 and Table 2 contain a comparison to a human expert baseline on the training datasets. We changed the discussion to highlight the human aspect.

---

### Decision · Program_Chairs · 2021-01-07
**Final Decision**

**Decision:**

Reject

**Comment:**

Paper was reviewed by four expert reviewers who identified the following pros/cons for the approach:

> Pros:
- Paper addresses an important problem [R3]
- Formulation is simple, elegant an easily adoptable [R2, R3, R4]
- Experimental results are compelling (ECCV challenge winner) [R2, R3, R4]

> Cons:
- Experiments  are using "novel" evaluation metric with little justification [R1, R3]
- Details of the approach are unclear [R1]
- Lack of simpler baselines in evaluation [R4]
- No comparison with handlabelling counterparts [R4]

Authors have addressed a number of comments in the rebuttal. With [R2] and [R4], generally, being convinced about accepting the paper.  However, [R1] and [R3] actually became less positive about the paper as a function of the rebuttal. [R3] mentioned reducing the score to a 3 and [R1] to a 6 in the discussion. It is unclear why these changes are not reflected in the publicly facing reviews. However, the fact that two of the four reviewers were disappointed with author responses and became LESS convinced about the paper is problematic in AC's opinion.

AC also agrees that standard performance metrics should have been included. It is obviously reasonable to propose new metrics, but doing so should be accompanied by (1) reporting of performance with the original standard metric(s) and (2) justification for where prior metrics are problematic or faulty. While the proposed metric is reasonable for the specific use case outlined in Appendix C, it doesn't preclude standard evaluation metrics nor points out why they would be inappropriate or faulty.

Overall, AC likes the paper and agrees that it presents a valuable approach that should be published. Unfortunately, the unjustified use of non-standard metrics without accompanied evidence, as noted above, is problematic and needs to be addressed in AC's opinion before that can happen. Since this issue was not addressed by authors in the rebuttal, AC sees no recourse but to reject the paper at this time, with a strong encouragement to address the aforementioned issue and to resubmit to CVPR or another top-tier upcoming venue.